# LARGE LANGUAGE MODELS AS TOOL MAKERS

**Tianle Cai**[1,2*]   **Xuezhi Wang**[1]   **Tengyu Ma**[1,3†]   **Xinyun Chen**[1]   **Denny Zhou**[1]
[1]Google Deepmind   [2]Princeton University   [3]Stanford University

## ABSTRACT

Recent research has highlighted the potential of large language models (LLMs) to improve their problem-solving capabilities with the aid of suitable *external tools*. In our work, we further advance this concept by introducing a *closed-loop framework*, referred to as LLMs As Tool Makers (LATM), where LLMs *create* their own reusable tools for problem-solving. Our approach consists of two phases: 1) tool making: an LLM acts as the *tool maker* that crafts *tools* for a set of tasks, where a tool is implemented as a Python utility function. 2) tool using: another LLM acts as the *tool user*, which applies the tool built by the tool maker for problem-solving. The tool user can be either the same or a different LLM from the tool maker. On the problem-solving server side, tool-making enables continual tool generation and caching as new requests emerge. This framework enables subsequent requests to access cached tools via their corresponding APIs, enhancing the efficiency of task resolution. Beyond enabling LLMs to create their own tools, our framework also uncovers intriguing opportunities to optimize the *serving cost* of LLMs: Recognizing that tool-making requires more sophisticated capabilities, we assign this task to a powerful, albeit resource-intensive, model. Conversely, the simpler tool-using phase is delegated to a lightweight model. This strategic division of labor allows the once-off cost of tool-making to be spread over multiple instances of tool-using, significantly reducing average costs while maintaining strong performance. Furthermore, our method offers a *functional cache* through the caching and reuse of tools, which stores the functionality of a class of requests instead of the natural language responses from LLMs, thus extending the applicability of the conventional cache mechanism. We evaluate our approach across various complex reasoning tasks, including Big-Bench tasks. With GPT-4 as the tool maker and GPT-3.5 as the tool user, LATM demonstrates performance equivalent to using GPT-4 for both roles, but with a significantly reduced inference cost. The codebase can be found in https://github.com/ctllll/LLM-ToolMaker.

## 1 INTRODUCTION

Large language models (LLMs) have demonstrated outstanding capabilities across a broad array of NLP tasks (Brown et al., 2020; Chowdhery et al., 2022; Zhang et al., 2022; Hoffmann et al., 2022; OpenAI, 2023; Google, 2023) and have even shown promising signs of achieving certain aspects of artificial general intelligence (Bubeck et al., 2023; Kosinski, 2023). Moreover, analogous to the evolution of human intelligence, recent research has unveiled the potential of augmenting LLMs with *external tools*, thereby significantly enhancing their problem-solving capacities and efficiencies (Yao et al., 2023; Liu et al., 2023; Parisi et al., 2022; Schick et al., 2023).

However, the applicability of these tool-using methods is largely contingent on the availability of suitable tools. According to the lessons learned from the evolutionary milestones of humans, a crucial turning point was that humans got the ability to fabricate their own tools to address emerging challenges. Inspired by the importance of tool-making for humans, in this work, we embark on an initial exploration to apply this evolutionary concept to the realm of LLMs. We propose a *closed-loop framework*, which we term as LLMs As Tool Makers (LATM), enables LLMs to generate their own reusable tools to tackle new tasks. Our approach comprises two key stages: 1) tool making: an LLM,

---

*Work done as a Student Researcher at Google Deepmind.

†Work done as a Visiting Researcher at Google Deepmind.

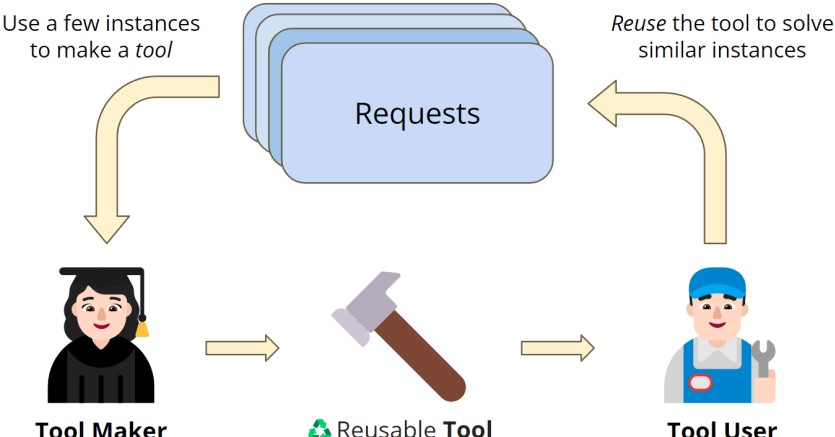

Figure 1: **The closed-loop framework of LATM.** In situations with numerous problem-solving requests, directly utilizing a powerful LLM to solve all the instances can result in high costs. On the other hand, lightweight models are cost-effective but usually struggle with complex tasks. LATM leverages the strengths of both models by employing a powerful model as the tool maker to generate reusable tools (implemented as Python functions) for tasks observed in the requests and pass the tool to a cost-effective tool user model for solving similar instances in the following requests. This approach allows the lightweight model to achieve performance comparable to the powerful model while maintaining greater cost-efficiency.

known as the *tool maker*, designs *tools* (implemented as Python functions) specifically for a given task. 2) tool using: another LLM referred to as the *tool user*, which can be the same as the tool maker, applies the tools to handle new requests. The two-stage design allows LATM to allocate jobs in each stage to the most suitable LLM. Specifically, the tool-making process, which requires a high degree of capability, can be assigned to a powerful albeit resource-intensive model (e.g., GPT-4). On the other hand, the tool-using process, which is comparatively simpler, can be assigned to a lightweight and cost-effective model (e.g., GPT-3.5 Turbo). This approach not only enhances the problem-solving capabilities of LLMs, but also significantly reduces the average computational cost of addressing a series of tasks.

As the tool-making process needs to be executed only once for a given functionality, the resulting tools can be reused across different task instances. This approach paves the way for a scalable and cost-efficient solution for handling complex task. For instance, consider a task where a user ask the LLM to schedule a meeting that works for everyone (e.g., in email conversations). Lightweight models like GPT-3.5 Turbo often struggle with such tasks that involve complex arithmetic reasoning. In contrast, more powerful models (e.g., GPT-4) can find the correct solutions, despite that the inference costs become much higher. LATM overcomes these hurdles by employing a powerful yet expensive model as the tool maker, and passing it to a cost-effective model as the tool user, for subsequent usage. After the tool has been forged, the lightweight tool user can use it to solve the task efficiently with high performance. This paradigm can similarly be applied to recurring tasks in various workflows, such as parsing and analyzing web documents into specific data formats or formulating routing plans that satisfy several custom requirements, or being used to solve popular games like the 24-game, Sudoku.

In the context of serving cost reduction, LATM introduces the opportunity of creating a *functional cache* for the LLM server. Specifically, consider a streaming setting where the LLM server continuously receives a sequence of requests. Traditional cache systems, such as GPTCache (Zilliz, 2023), store the responses generated by the LLMs and reuse them for *textually* similar requests. However, with the capacity for tool-making that LATM introduces, the system can store tools crafted by the tool maker and reuse them for *functionally* analogous requests. This novel approach, combined with the strategic division of labor between the tool maker and tool user, has the potential to considerably reduce the average cost of serving a sequence of requests while maintaining high performance.

Our experiments validate the effectiveness of this approach on a range of complex reasoning tasks, including several challenging Big-Bench tasks (Srivastava et al., 2022). The results show that LATM can achieve performance on par with more resource-intensive models while being more cost-effective. This novel approach to LLMs, which mimics the evolutionary leap of humans in creating and using tools, opens up exciting possibilities for a growing community with LLM-generated tools.

## 2 RELATED WORK

**Chain of thought (CoT).**    Recently, significant progress has been made in enhancing the problem-solving abilities of large language models (LLMs) for complex tasks. For instance, CoT prompting (Wei et al., 2022; Wang et al., 2022) has been proposed to bolster LLM reasoning capabilities, demonstrating improved performance across various reasoning and natural language processing tasks. CoT is typically articulated through natural languages (Ling et al., 2017; Cobbe et al., 2021; Suzgun et al., 2022; Shi et al., 2022; Zhou et al., 2022), yet it might also be effectively represented using programming languages (Amini et al., 2019; Austin et al., 2021; Nye et al., 2021; Chowdhery et al., 2022; Gao et al., 2023; Chen et al., 2022). More recently, Arora et al. (2023) proposed using LLMs to generate structured views over documents, balancing quality and cost by ensembling extractions from multiple synthesized functions. Our method shares a similar spirit with Arora et al. (2023) in managing cost and quality trade-offs but focuses on more general use cases.

**Augmenting language models with tools.**    Recent works have explored the potential of using external tools to supplement LLMs' capabilities for complex tasks. Yao et al. (2023); Yang et al. (2023) proposed augmenting reasoning traces with task-specific actions in LLMs, enabling models to reason and act synergistically. Various studies (Liu et al., 2023; Parisi et al., 2022; Schick et al., 2023; Shen et al., 2023; Lu et al., 2023; Paranjape et al., 2023; Liang et al., 2023) have demonstrated that supplementing LLMs with tools, such as calculators, search engines, translation systems, calendars, or even API calls on other models, can help solve tasks that are not easily addressed by LLMs alone.

Similar to LATM, methods like Chameleon (Lu et al., 2023) also incorporate Python executors in the pipeline. However, their primary focus is on using Python executors to accurately solve sub-steps involving arithmetic reasoning, similar to Gao et al. (2023); Chen et al. (2022). In contrast, we use Python executors to create *reusable* tools for addressing other task instances. Furthermore, the separation of the *tool maker* and *tool user* enables the use of a lightweight model for most inferences, thus enhancing efficiency and cost-effectiveness in LATM.

**Adaptive generation in language models.**    In addition, recent research has proposed methods to adaptively control decoding in LLMs to improve text generation efficiency (Leviathan et al., 2022; Chen et al., 2023a; Xia et al., 2023). Speculative decoding is based on the notion that generating text tokens (a more expensive process) can be expedited with a faster yet less powerful model while approximating the performance of larger, costlier models by using them to score generated tokens (a much faster process). Our approach of passing tools from a more expensive model to a smaller, faster model also shares a similar spirit of adaptive computing. Instead of altering the decoding procedure, we transfer newly generated tools between models to boost both the performance and efficiency of an LLM in solving tasks.

**Language model cascades.**    There is recent evidence that LLMs can enable repeated interactions and that multiple LLMs can be combined to extend their capabilities further (Wu et al., 2022; Zhou et al., 2022; Dohan et al., 2022; Chen et al., 2023c). Also, Chen et al. (2023b) demonstrated that identifying optimal LLM combinations can help reduce costs while improving accuracy. Our motivation aligns with these findings; however, rather than merely cascading LLMs, we identify task categories that can be better addressed using new tools generated by a larger model and assign each individual inference within that task category to a smaller model.

**Early attempts on tool-making.**    Concurrent and independent to our work, several early attempts have been made towards using LLMs to make tools. Wang et al. (2023) conducted research within the Minecraft environment and demonstrated the ability of an LLM-powered agent to acquire new skills in the form of programs. Similarly, Qian et al. (2023) proposes a method of decomposing problem-solving for each individual instance into an abstract tool creation phase and a concrete tool

application phase. Our work aligns with the spirit of both Wang et al. (2023) and Qian et al. (2023) in the aim to let LLMs to generate their own tools for problem-solving. However, we also underscore the significance of tool reusability and cost-effectiveness stemming from the division of labor. The idea of tool making is also mentioned in a recent survey paper (Qin et al., 2023).

## 3 LLM AS TOOL MAKER (LATM)

**Tool making template (One-time 🚀 )**

**Tool proposing**: Write a generic Python function (the Tool) to solve three training samples.
**Tool verification**: Write unit tests to convert three validation samples into function call and validate the correctness.
**Tool wrapping**: Gather the function from the proposing stage and the examples of how to convert problems to function calls from the verification stage into a reusable **Wrapped Tool**.

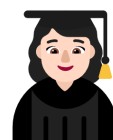

**Tool Maker** (e.g., GPT-4): Strong performance but slow and expensive 🪖

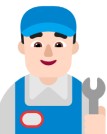

**Tool User** (e.g., GPT-3.5 Turbo): Weaker performance but much faster and cheaper 🐰

**Tool using template (Reusable ♻ )**

Convert problem into function call according to the **Wrapped Tool**

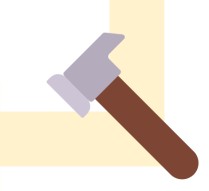

♻ **Wrapped Tool**

Figure 2: **The pipeline of LATM.** LATM can be divided into two stages: 1) tool making: a powerful yet more expensive model serves as the tool maker to generate generic and reusable tools from a few demonstrations; 2) tool using: a lightweight and cheaper model serves as the tool user to use the tool to solve various instances of the task. The tool-making stage can be further divided into three sub-stages: (i) tool proposing: the tool maker makes an attempt to generate the tool (Python function) from a few training demonstrations, if the tool is not executable, report the error and generate a new one (fix the function); (ii) tool verification: the tool maker runs unit tests on validation samples, if the tool does not pass the tests, report the error and generate new tests (fix the function calls in unit tests); and (iii) tool wrapping: wrapping up the function code and the demonstrations of how to convert a question into a function call from unit tests, preparing usable tools for tool user.

### 3.1 MAKING NEW TOOLS AND REUSE THEM

In the LATM paradigm, the main process can be split into two stages: Tool Making and Tool Using. Each stage utilizes different types of Large Language Models (LLMs) to balance performance and cost-effectiveness. All the prompts used in our experiments are shown in Appendix C.

**Tool Making.**    This stage employs a powerful yet more expensive model, such as GPT-4, to serve as the *tool maker*. *Tool maker*'s role is to create a generic and reusable *tool* (implemented as a Python function) from a few demonstrations of a task. This stage can be further divided into three sub-stages:

- **Tool Proposing:** In this stage, *tool maker* attempts to generate a Python function to solve the demonstrations from the given task. This process follows the "programming by example" (PbE)

paradigm (Halbert, 1984) where several concrete demonstrations are provided, and the model is required to write programs that produce the demonstrated behaviors. In our experiments, we use 3 demonstrations for this stage. If the proposed tool is unexecutable or encounters errors, *tool maker* appends the error messages to the history and makes another attempt.

- **Tool Verification:** In this stage, the *tool maker* generates unit tests using validation samples and subsequently executes these tests on the proposed tool. We utilize 3 validation samples in our experiments. If the tool fails any of these tests, the *tool maker* records the error in its history and makes an attempt to rectify the issues within the unit tests (this procedure will only correct the function calls in the unit test part and will not correct the function). The ability of LLMs to self-debug has been demonstrated effectively in recent research (Madaan et al., 2023; Chen et al., 2023c; Lu et al., 2023; Kim et al., 2023). However, within the LATM pipeline, the verification stage serves a slightly different usage. This stage fulfills two key roles: 1) it provides examples that demonstrate how to convert natural language questions into function calls, and 2) it verifies the tool's reliability, enabling the entire process to be fully automated.

- **Tool Wrapping:** If the execution or verification fails over a preset threshold, the Tool Making stage is viewed as failed. Otherwise, *tool maker* is ready to prepare the *wrapped tool* for *tool user*. This step involves wrapping up the function code and providing demonstrations of how to convert a task into a function call. These demonstrations are extracted from the Tool Verification step, which converts questions into unit tests. This final product is then ready for use by the *tool user*. Please see Appendix D for examples of the wrapped tools.

**Tool Using.** This second stage involves a lightweight and cost-effective model, such as GPT-3.5 Turbo, to serve as the *tool user*. The *tool user*'s role is to utilize the verified tool to solve various instances of the task. The prompt for this stage is the *wrapped tool* which contains the function for solving the task and demonstrations of how to convert a task query into a function call. With the demonstrations, *tool user* can then generate the required function call in an *in-context learning* fashion. The function calls are then executed to solve the task. Optionally, postprocessing can be applied to convert the output to match the required format of the task, such as options for multiple-choice questions.

The tool-making stage, including tool proposing, verification, and wrapping, only needs to be performed *once* for each type of task. The resulting tools can then be *reused for all instances* of that task. This makes LATM significantly more efficient and cost-effective than using a powerful model alone. Furthermore, the Python function tools are a more generic form of Chain-of-Thought, enhancing the overall utility and flexibility of the LLMs, as they can be used to solve questions that involve algorithmic reasoning ability (Veličković and Blundell, 2021).

## 4 LATM FOSTERS A FUNCTIONAL CACHE MECHANISM FOR LLM SERVING

In real-world scenarios, tasks often arrive in a sequential stream. To address this, we introduce a third LLM, the *dispatcher*, that decides whether to engage the *tool user* or *tool maker* for each incoming task. While this tool selection function mirrors existing works (Lu et al., 2023; Shen et al., 2023; Schick et al., 2023; Paranjape et al., 2023), our *dispatcher* distinctively contributes to creating a *functional cache*—it discerns new tasks that cannot be resolved with existing tools, thereby triggering the *tool maker* to generate appropriate tools for these tasks.

The *dispatcher* maintains a repository of existing tools crafted by the *tool maker* in the format of function APIs. Upon receipt of a new task instance, the *dispatcher* first attempts to locate a compatible tool within the cache. If such a tool is present, the *dispatcher* assigns the instance and corresponding tool to the *tool user* for resolution. However, if no suitable tool is available, the *dispatcher* identifies this as a novel task, either solving it with a powerful model or, if necessary, invoking a human labeler. These new instances are then cached until a sufficient number are amassed to craft a new tool, further enriching the *functional cache*. This mechanism allows for the *functionally* similar tasks to reuse these tools, expanding the coverage of the classic cache mechanism and reducing the overall serving cost. Given the simplicity of the dispatching task, a lightweight model equipped with appropriate prompts (See Appendix C) can efficiently serve as the *dispatcher*, adding only a marginal cost to the entire pipeline.

Figure 3: **An illustration of the Tool Proposing and Tool Using stages of the LATM pipeline for the Logical Deduction task (Srivastava et al., 2022)**. This task requires determining the order of five objects based on several given conditions. In the Tool Proposing stage, the *tool maker* (such as GPT-4) formulates a generic Python function capable of solving the provided $k$ demonstrations from the task (where $k$ equals 3 in our experiments). The *tool maker* generates a search algorithm that enumerates all possible orderings and verifies each against the provided conditions. During the tool-using stage, the *tool user* translates each natural language question into a series of conditions, generating function calls to utilize the tool for each task instance.

## 5 EXPERIMENTS

### 5.1 EXPERIMENTAL SETUP

**Datasets.** We evaluate our approach on six datasets from diverse domains, including Logical Deduction, Tracking Shuffled Objects, Dyck Language, Word Sorting, Chinese Remainder Theorem, and Scheduling Meeting. The first five datasets are sourced from BigBench (Srivastava et al., 2022). We take the 5 objects version of the Logical Deduction and Tracking Shuffled Objects tasks, referred to as Logical Deduction (5) and Tracking Shuffled Objects (5) in the paper. We also constructed the Scheduling Meeting task to demonstrate the effectiveness of LATM in real-world scenarios. Detailed information on dataset generation can be found in Appendix E. We divide each dataset into training, validation, and test sets, containing 3, 3, and 240 instances, respectively.

**Model settings.** During the tool-making stage, we set the temperature to 0.3 to introduce randomness to the generation process, allowing for retries if necessary. For this stage, we conduct experiments using GPT-4 and GPT-3.5 Turbo models with the `ChatCompletion` API, always appending the response to the chat history to create an interactive experience. In the tool-using stage, the LLM API call is made only once, and we also perform ablation studies on GPT-3-type models with the standard `Completion` API. When using the tools, we consistently set the temperature to 0.0. We set the maximal retry times to be 3 for the tool-proposing and tool-verification stages.

### 5.2 EFFECTIVENESS OF THE TOOL-MAKING STAGE

In the tool-making stage, we use a powerful yet slower model to generate *generic* Python functions tailored to a specific task. This step is performed only *once* for each task, and the overhead is amortized across all instances of that task. In our experiments, we use GPT-4 (OpenAI, 2023) as a

| Logical Deduction (5) | Tracking Shuffled Objects (5) | Dyck Language | Word Sorting | Chinese Remainder Theorem | Schedule Meeting |
|---|---|---|---|---|---|
| Search | Simulation | Stack | Sort | Search/Extended Euclidean | Interval intersections |

Table 1: **The utility functions generated by *tool maker* to solve the tasks.**

representative *tool maker*, while we explore other models' tool-making capabilities in Section 5.5. We provide several few-shot exemplars for the language model, guiding it to generate generic Python programs, as illustrated in Figure 3.

Our observations indicate that when GPT-4 is employed as the *tool maker*, the model frequently devises suitable algorithms for solving tasks. For instance, as shown in Table 1, the *tool maker* creates code to solve the logical deduction task by searching through all permutations and selecting the correct one that satisfies the given constraints. In our experiment, the tool-verification stage is mainly used to provide examples that demonstrate how to convert natural language questions into function calls, and we only observe 2 cases out of the 60 trials that the *tool maker* can correct its mistakes with the guide of error messages. See Section 5.5 for more discussions on the *tool maker*.

## 5.3 LATM IMPROVES THE PERFORMANCE OF LIGHTWEIGHT LLMs

In Table 2, we compare the performance of Chain-of-Thought prompting (Wei et al., 2022) with our method, LATM. We employ GPT-4 as the *tool maker* to generate tools for the six tasks, and evaluate the performance of both GPT-3.5 Turbo and GPT-4 as *tool user*. The results demonstrate that with the help of the *tool*, a lightweight model like GPT-3.5 Turbo can achieve performance on par with GPT-4, significantly outperforming CoT prompting. Additionally, the average cost of using GPT-3.5 Turbo with the *tool* is much lower compared to using GPT-4. This highlights the effectiveness of LATM in enhancing the performance of lightweight models and therefore reducing the cost compared to employing expensive models. Intriguingly, for the Dyck Language task, GPT-3.5 Turbo as the *tool user* even surpasses GPT-4 in its role as the *tool user*. Upon investigating the failure cases, we find that when converting the question into a function call, GPT-4 occasionally superfluously closes some brackets within the argument instead of leaving the argument unchanged and letting the function solve it, which leads to incorrect function output.

| *Tool User* Model | Method | Logical Deduction (5) | Tracking Shuffled Objects (5) | Dyck Language | Word Sorting | Chinese Remainder Theorem | Schedule Meeting | Cost on $n$ samples |
|---|---|---|---|---|---|---|---|---|
| GPT-3.5 Turbo | CoT | 66.4 | 61.6 | 20.4 | 59.2 | 0.0 | 18.9 | $O(nc)$ |
| | LATM | 79.7 (+13.3) | 99.6 (+38.0) | **92.2** (+71.8) | 98.3 (+39.1) | **100.0** (+100.0) | **100.0** (+81.1) | $O(nc + C)$ |
| GPT-4 | CoT | **88.8** | **100.0** | 63.6 | 90.9 | 0.0 | 55.6 | $O(nC)$ |
| | LATM | 86.6 | **100.0** | 87.5 | **99.1** | **100.0** | **100.0** | $O(nC)$ |

Table 2: **Accuracy comparison between LATM and Chain-of-Thought.** The six tasks are detailed in Section 5.1. For LATM, the *tool* is created by GPT-4 and utilized by both GPT-3.5 Turbo and GPT-4. The results demonstrate that the application of LATM can significantly enhance the performance of GPT-3.5 Turbo, often surpassing or matching GPT-4's performance with CoT in certain scenarios. The last column depicts the overall cost of processing $n$ samples. Here, $C$ represents the cost of one call to GPT-4, while $c$ denotes the cost of one call to GPT-3.5 Turbo. At the time of writing this paper, $C$ is over 15x larger than $c$. The few-shot CoT demonstrations for the first four tasks are provided by Suzgun et al. (2022), while for the last two tasks, we apply direct few-shot prompting without CoT.

## 5.4 ADAPTING LATM TO A DYNAMIC STREAM OF DIVERSE TASKS

As discussed in Section 4, we can adapt LATM to handle a dynamic stream where instances from potentially different tasks emerge in real-time. In this setting, we introduce an additional model, the *dispatcher*, tasked with identifying the task to which each incoming instance pertains. We employ GPT-3.5 Turbo for this role, evaluating its effectiveness in two key functions: 1) Identifying and employing existing tools from the *functional cache* to resolve an incoming instance, and 2)

| *Tool Maker* Model | Logical Deduction (5) | Tracking Shuffled Objects (5) | Dyck Language | Word Sorting | Chinese Remainder Theorem | Schedule Meeting |
|---|---|---|---|---|---|---|
| GPT-3.5 Turbo | 0/5 | 0/5 | 5/5 | 5/5 | 5/5 | 0/5 |
| GPT-4 | 3/5 | 4/5 | 5/5 | 5/5 | 5/5 | 3/5 |

Table 3: **Success rate of generating new tools (Python functions that pass the tool-verification step) in the tool-making stage with GPT-4 v.s. GPT-3.5 Turbo.** We run 5 trials for each model on each task, $n/5$ means $n$ trails out of 5 successes to produce a valid tool. For hard tasks like Logical Deduction and Tracking Shuffled Objects, GPT-3.5 Turbo fails in all trials, showing the necessity of using a more powerful model as *tool maker*.

Detecting unseen tasks and triggering the *tool maker* to create appropriate tools for these tasks. This experimental setup helps assess how effectively our system can reduce serving costs by reusing and extending the *functional cache* in a dynamic, multi-tasking scenario.

**Identifying existing tools.** The first part of our evaluation assesses the *dispatcher*'s capability to recognize existing tools within the *functional cache* that correspond to a given instance, analogous to the cache fetching phase of traditional cache systems. To this end, we generate a test set of 100 samples, randomly mixed from the six tasks discussed in Section 5.1. For each instance, the *dispatcher* is tasked to determine the appropriate tool from existing ones, utilizing prompts containing task examples associated with these tools (See Appendix C). Success is measured by the correct identification of the tool. Over five random constructions of the test set, the accuracy in correctly determining the suitable tool is $95\% \pm 2\%$.

**Requesting tool-making.** The second part of our evaluation tests the *dispatcher*'s ability to request tool-making for instances originating from an unseen task. This situation is akin to enqueuing a new instance into the cache when a cache miss happens. We randomly designate four tasks as existing tasks with readily available tools and select four other tasks for testing—two of these are unseen, and the other two fall within the realm of existing tasks. Again, a test set of 100 samples is generated. For each instance in the test set, the *dispatcher* determines whether it needs to request tool-making or if an existing tool can solve the instance. Over multiple runs, the accuracy of making the correct decision stands at $96\% \pm 3\%$, demonstrating the robustness of our approach in efficiently managing the *functional cache*.

The above results illustrate that the *dispatcher* can effectively recognize existing tools and accurately request tool-making for unseen tasks, all while maintaining high performance. These findings highlight the potential of LATM to be seamlessly adapted to a streaming environment encompassing a diverse range of tasks. This validation serves to fortify the viability of our framework in real-world applications, particularly where the efficient management of *functional cache* is paramount.

## 5.5 ABLATION STUDY

**Capacity required for the tool-making language model.** We investigate the capacity requirements for the language model used in the tool-making stage (See Table 3). Generally, we found that a more powerful and expensive model better serves the purpose, as this stage is performed only once for each task, and high accuracy is crucial for effectively passing tools to a smaller model. Specifically, on hard tasks like Logical Deduction and Tracking Shuffled Objects, GPT-3.5 Turbo fails in all the 5 trails. And the major failure reason is that the tool is not general enough and may only work on the training samples. On the other hand, we also discovered that for easy tasks, the *tool maker* can be a lightweight language model. For simple tasks like Word Sorting, GPT-3.5 Turbo can effortlessly generate a program that solves the task. Another limitation that may contribute to the *tool maker*'s failure is the context length constraints. Since we use the entire history in each step of tool-making to enhance the reliability of the tool-making stage, this also introduces a longer context. In this case GPT-4 with 8192 context length is preferable.

**Capacity required for the tool-using language model.** In this section, we investigate the capacity requirements for the tool-using model. The results are presented in Table 4. We observed that GPT-3.5

|  | GPT-3.5 Turbo | text-davinci-002 | davinci | curie | babbage | ada |
|---|---|---|---|---|---|---|
| Logical Deduction (5) | **79.7%** | 58.2% | 11.6% | 6.5% | 11.6% | 3.0% |
| Tracking Shuffled Objects (5) | 99.6% | **100.0%** | 62.1% | 20.7% | 16.4% | 5.2% |
| Dyck Language | **92.2%** | 35.8% | 16.4% | 18.1% | 9.1% | 9.9% |
| Word Sorting | **98.3%** | 60.8% | 26.6% | 7.3% | 7.3% | 0.9% |
| Chinese Remainder Theorem | **100.0%** | **100.0%** | 99.6% | 93.1% | 75.0% | 66.0% |
| Schedule Meeting | **100.0%** | **100.0%** | 62.9% | 59.1% | 23.2% | 0.0% |
| Cost ($ per 1K tokens) | 0.002 | 0.02 | 0.02 | 0.002 | 0.0005 | 0.0004 |

Table 4: **A performance comparison of various *tool user* models, all using the same *tool* generated by GPT-4.** All costs are based on the rates at the time of writing. Of all the models, GPT-3.5 Turbo demonstrates the best trade-off between performance and cost. We opted for GPT-3 models *prior* to instruction tuning (ada instead of text-ada-001, etc.), as we observed that the models after instruction tuning underperformed in the tool-using stage. We postulate that this is due to the instruction tuning impairing the in-context learning ability, which is essential for the tool-using stage.

Turbo offers the best balance between performance and cost among all the models tested. Regarding the older GPT-3 series of models (ada, babbage, curie, davinci), we found that models that *before* instruction tuning often perform better than their counterparts post instruction tuning (text-ada-001, etc.). We hypothesize that the instruction tuning phase in these models may adversely impact the in-context learning ability, which is crucial for the tool-using stage.

**CoT as a tool does not help.**  In addition to LATM, we investigate if we can improve task performance by reusing Chain-of-Thought (CoT) from a larger model to a smaller model similar to LATM pipeline. Specifically, we use the same larger model (GPT-4) in the "CoT-making" stage, using zero-shot prompting "Let's think step by step." to elicit the intermediate thought steps, and then use the generated CoT to the same smaller tool-using model (GPT-3.5 Turbo). We test this on two tasks and report the results Table 5. We observe that using CoT from a large model has a similar or even worse performance than human-written CoT, which is much worse than LATM.

| Accuracy | GPT-4 CoT | Human-written CoT | LATM |
|---|---|---|---|
| Logical Deduction (5) | 36.8 | 66.4 | **79.7** |
| Tracking Shuffled Objects (5) | 63.2 | 61.6 | **99.6** |

Table 5: **Accuracy of using CoT generated by GPT-4.** The performance is similar to human-written CoT, which is much worse than LATM.

## 6  CONCLUSION AND FUTURE WORK

We introduced LATM, a closed-loop framework empowering large language models (LLMs) to create and utilize their own tools for diverse tasks. Our approach, inspired by human's evolutionary strides in tool creation, employs two key stages: Tool Making and Tool Using. This division of labor allows us to harness the capabilities of advanced LLMs while significantly reducing computational costs. Our experiments confirmed the efficacy of LATM across various complex tasks, demonstrating that our framework performs comparably to resource-intensive models while being more cost-effective. In addition, we show that adding another *dispatcher* LLM can further provide flexibility to our framework, enabling on-the-fly tool creation and usage.

In our evaluation process, we identified a significant lack of high-quality datasets that authentically represent daily human-computer interactions, including recurring tasks such as scheduling meetings or booking flights over email or phone calls, in their raw natural language format. We anticipate that our work will stimulate the research community to create such datasets, which could prove instrumental in cultivating the next generation of AI systems. These systems, capable of generating and applying their own tools, will be equipped to tackle complex tasks more effectively. An exciting avenue for future research is enabling the *tool maker* to refine and upgrade existing tools to manage new problem instances, much like in software development. This adaptability could further catalyze the evolution of the AI ecosystem, unlocking a wealth of opportunities.

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

## A    ILLUSTRATION OF THE DISPATCHER

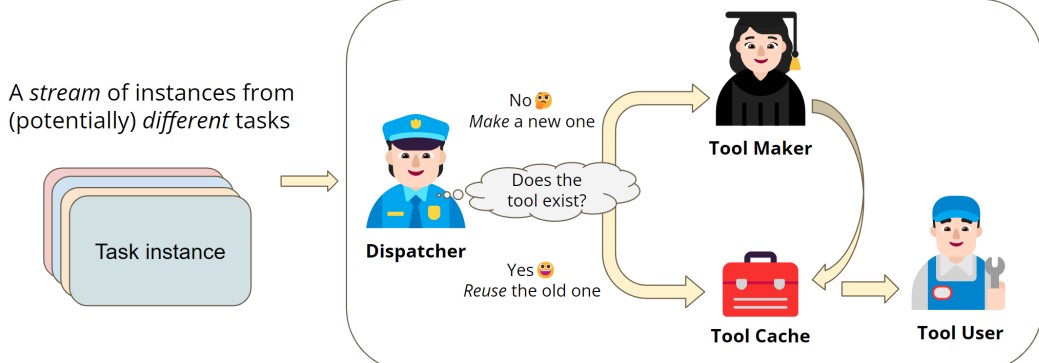

Figure 4: **An illustration of the Dispatcher that enables functional cache mechanism**. In an online setting where task instances arrive sequentially, the *dispatcher*, a lightweight model, assesses each incoming instance. If a suitable tool already exists in the cache to tackle the task, the *dispatcher* selects this tool and forwards the task instance to the *tool user* for resolution. If no suitable tool is found, the *dispatcher* routes the task instance to the *tool maker* to create a new tool that can be used by *tool user* later.

## B    BROADER IMPACT AND LIMITATIONS

This paper explores the potential of enabling Large Language Models (LLMs) to create their own tools, thus allowing them greater autonomy in developing their ecosystem. While this avenue of research is promising, it also raises important ethical, safety, and control considerations that need to be carefully addressed.

One of the most significant impacts of our work lies in the potential for LLMs to grow and achieve unprecedented capabilities automatically. This could significantly enhance the range and complexity of tasks these models can handle, potentially revolutionizing fields such as customer service, technical support, and even areas of research and development. It could lead to more efficient use of computational resources and a reduction in human intervention, especially for routine or repetitive tasks.

However, this newfound autonomy of LLMs is a double-edged sword. As we endow LLMs with the ability to generate their own tools, we also create a scenario where the quality of the tools they develop may not always meet the standards or expectations set by human developers. Without proper safeguards, there's a risk that these models could generate solutions that are suboptimal, incorrect, or even potentially harmful. Furthermore, as LLMs become more autonomous, the potential for loss of control increases. If these models are widely used without appropriate regulation, there could be unforeseen consequences, potentially even leading to scenarios where humans lose control over the AI systems.

In this study, we have not addressed these control and safety issues in depth, and our work has some limitations. Our proposed framework, **LLM As Tool Maker**, while effective in the tested scenarios, is still in its early stages of development. It is crucial to note that the real-world performance and safety of the system may vary based on the complexity and nature of the tasks it is applied to. Additionally, the evaluation and validation of the tools created by the *tool maker* in a real-world setting is a challenge that needs to be addressed.

## C  LATM PROMPTS

---

**Tool Maker Prompt**

```
Please write a generic Python function to solve this type of
↪  problems using only standard python libraries. The output
↪  of the function can later be converted to the answer
↪  (option for multiple choice question). All the function
↪  should be wrapped by
```python
```
```

---

**Tool Verifier Prompt**

```
Write unit tests to verify the correctness of the function on
↪  the questions above using the following format:
```python
{parse the question into the arguments of the function}
{call the function and save the return value in a variable
↪  named "ret"}
{for multiple choice question, parse the options}
{convert the return value "ret" to the answer (if the
↪  question is a multiple choice question, convert to an
↪  option) and save it in a variable named "ans", otherwise}
{assert ans == the provided answer (if the question is a
↪  multiple choice question, assert ans == option)}
```
```

---

**Tool Wrapper Prompt**

```
Success! The function is correct. We will need to summarize
↪  the function and use cases up for further use. Please
↪  extract the information from the history in the following
↪  format:

Here is a function to solve a class of problems:
```python
{the function, including necessary imports}
```

Use cases:
Question: {question (including options)}
Solution:
```python
{parse the question into the arguments of the function}
{call the function and save the return value in a variable
↪  named "ret"}
{for multiple choice question, parse the options}
{convert the return value "ret" to the answer (if the
↪  question is a multiple choice question, convert to an
↪  option) and save it in a variable named "ans", otherwise}
```
Do this for all the questions in the verification step.
```

**Dispatcher Prompt**

```
Here are several functions that can be used to solve some
↪  task:

Task: logical_deduction_five_objects

API: find_order(objects, constraints):
Finds the order of objects that satisfies a given set of
↪  constraints.
objects: A list of unique objects (strings) to be ordered.
constraints: A list of lambda functions that represent the
↪  constraints on the order of objects. Each constraint
↪  should take the order of objects as input and return a
↪  boolean value (True if the constraint is satisfied, False
↪  otherwise).
return: A tuple representing the order of objects that
↪  satisfies all the constraints. If no such order exists,
↪  the function returns None.

===

Task: tracking_shuffled_objects_five_objects

API: square_dance(initial_partners, switches):
This function takes an initial list of pairs and a list of
↪  switches, and returns a dictionary representing the final
↪  state of the pairs after performing the switches.

initial_partners: A list of tuples, where each tuple contains
↪  two elements representing a pair (e.g., [("Alice",
↪  "goalkeeper"), ("Bob", "left midfielder"), ...]). The
↪  elements can be any type (e.g., strings, integers, etc.).

switches: A list of tuples, where each tuple contains two
↪  elements representing a pair of elements from the
↪  initial_partners list that will be switched (e.g.,
↪  [("Alice", "Claire"), ("Alice", "Bob"), ...]). The
↪  elements should match the types used in the
↪  initial_partners list.

return: A dictionary representing the final state of the
↪  pairs after performing the switches. The keys are the
↪  first elements of the pairs in the initial_partners list,
↪  and the values are the corresponding second elements
↪  after performing the switches (e.g., {"Alice": "right
↪  winger", "Bob": "center midfielder", ...}).

===

Skip other tasks

Here is a question:\n{question}\n\nAccoding to the API
↪  documents above, you may find some functions that can be
↪  used to solve the task, or, sometimes there does not
↪  exist proper function to solve the task. Figure out if
↪  there is function to solve the task and reply in the
↪  format:\nTask: {{task}} (reply unknown if no function can
↪  solve the question)
```

# D WRAPPED TOOLS

**Tool for Logical Deduction**

Here is a function to solve a class of problems:

```python
from itertools import permutations

def find_order(objects, constraints):
    for order in permutations(objects):
        valid = True
        for constraint in constraints:
            if not constraint(order):
                valid = False
                break
        if valid:
            return order
```

Use cases:

Question: The following paragraphs each describe a set of
↪ five objects arranged in a fixed order. The statements
↪ are logically consistent within each paragraph. On a
↪ shelf, there are five books: a white book, a green book,
↪ a brown book, a gray book, and an orange book. The gray
↪ book is to the right of the orange book. The green book
↪ is the second from the right. The brown book is to the
↪ right of the white book. The brown book is to the left of
↪ the orange book.
Options:
(A) The white book is the third from the left
(B) The green book is the third from the left
(C) The brown book is the third from the left
(D) The gray book is the third from the left
(E) The orange book is the third from the left
Solution:

```python
objects = ["white", "green", "brown", "gray", "orange"]

constraints = [
    lambda order: order.index("gray") >
    ↪ order.index("orange"),
    lambda order: order.index("green") == len(order) - 2,
    lambda order: order.index("brown") >
    ↪ order.index("white"),
    lambda order: order.index("brown") <
    ↪ order.index("orange")
]

ret = find_order(objects, constraints)
options = {
    "A": "white",
    "B": "green",
    "C": "brown",
    "D": "gray",
    "E": "orange"
}
ans = [k for k, v in options.items() if v == ret[2]][0]
```

Skip two more questions...

> **Tool for Tracking Shuffled Objects**
>
> Here is a function to solve a class of problems:
>
> ```python
> def square_dance(initial_partners, switches):
>     # Create a dictionary to store the current partners
>     current_partners = dict(initial_partners)
>
>     # Iterate through the switches and update the current
>     ↪   partners
>     for switch in switches:
>         dancer1, dancer2 = switch
>         partner1 = current_partners[dancer1]
>         partner2 = current_partners[dancer2]
>
>         # Swap the partners
>         current_partners[dancer1] = partner2
>         current_partners[dancer2] = partner1
>
>     return current_partners
> ```
>
> Use cases:
>
> Question: Alice, Bob, Claire, Dave, and Eve are on the same
> ↪   team in a soccer match. At the start of the match, they
> ↪   are each assigned to a position: Alice is playing
> ↪   goalkeeper, Bob is playing left midfielder, Claire is
> ↪   playing right winger, Dave is playing striker, and Eve is
> ↪   playing center midfielder.
> As the game progresses, pairs of players occasionally swap
> ↪   positions. First, Alice and Claire trade positions. Then,
> ↪   Alice and Bob trade positions. Then, Dave and Bob trade
> ↪   positions. Then, Bob and Eve trade positions. Finally,
> ↪   Dave and Eve trade positions. At the end of the match,
> ↪   Eve is playing
> Options:
> (A) goalkeeper
> (B) left midfielder
> (C) right winger
> (D) striker
> (E) center midfielder
> Answer: (C)
>
> Solution:
> ```python
> initial_positions = [("Alice", "goalkeeper"), ("Bob", "left
> ↪   midfielder"), ("Claire", "right winger"), ("Dave",
> ↪   "striker"), ("Eve", "center midfielder")]
> switches = [("Alice", "Claire"), ("Alice", "Bob"), ("Dave",
> ↪   "Bob"), ("Bob", "Eve"), ("Dave", "Eve")]
>
> ret = square_dance(initial_positions, switches)
> options = ["goalkeeper", "left midfielder", "right winger",
> ↪   "striker", "center midfielder"]
> ans = options.index(ret["Eve"]) + 1  # Convert the return
> ↪   value to an option index (1-based)
> ```
>
> Skip two more questions...

---

**Tool for Dyck Language**

```
Here is a function to solve a class of problems:
```

```python
def complete_sequence(input_str):
    stack = []
    closing_map = {'(': ')', '[': ']', '<': '>', '{': '}'}
    result = []

    for char in input_str:
        if char in closing_map.keys():
            stack.append(char)
        elif char in closing_map.values():
            if stack and closing_map[stack[-1]] == char:
                stack.pop()
            else:
                return "Invalid sequence"
        else:
            return "Invalid character"

    while stack:
        result.append(closing_map[stack[-1]])
        stack.pop()

    return ''.join(result)
```

```
Use cases:

Question: Complete the rest of the sequence, making sure that
↪  the parentheses are closed properly. Input:
↪  ([[[{}]]{<[<[{}]>]>}
Answer: ])
```

```
Solution:
```
```python
input_str = "([[[{}]]{<[<[{}]>]>}"
ret = complete_sequence(input_str)
ans = ret
```
```
Skip two more questions...
```

---

**Tool for Word Sorting**

```
Here is a function to solve a class of problems:
```python
def sort_words_alphabetically(word_list):
    return sorted(word_list)
```

Use cases:

Question: Sort the following words alphabetically: List:
↪   conference apparition ignore dutton layperson coupe
↪   superstitious westward turnoff messenger copra floruit
↪   primitive implement
Answer: apparition conference copra coupe dutton floruit
↪   ignore implement layperson messenger primitive
↪   superstitious turnoff westward

Solution:
```python
words1 = ["conference", "apparition", "ignore", "dutton",
↪   "layperson", "coupe", "superstitious", "westward",
↪   "turnoff", "messenger", "copra", "floruit", "primitive",
↪   "implement"]
ret1 = sort_words_alphabetically(words1)
ans1 = " ".join(ret1)
```

Skip two more questions...
```

> **Tool for Chinese Remainder Theorem**
>
> ```
> Here is a function to solve a class of problems:
> ```
>
> ```python
> def find_number(max_limit, divisors, remainders):
>     for num in range(max_limit + 1):
>         if all((num - remainder) % divisor == 0 for divisor,
>         ↪   remainder in zip(divisors, remainders)):
>             return num
>     return None
> ```
>
> ```
> Use cases:
> ```
>
> ```
> Question: There is a basket of no more than 1188877 durians.
> ↪   If we divide them equally among 41 penguins, we have 17
> ↪   left; if we divide them equally among 107 dinosaurs, we
> ↪   have 42 left; if we divide them equally among 271
> ↪   elephants, we have 260 left. How many durians are in the
> ↪   basket?
> ```
>
> ```
> Solution:
> ```
> ```python
> max_limit = 1188877
> divisors = [41, 107, 271]
> remainders = [17, 42, 260]
> ret = find_number(max_limit, divisors, remainders)
> ans = ret
> ```
> ```
> Skip two more questions...
> ```

> **Tool for Schedule Meeting**
>
> ```
> Here is a function to solve a class of problems:
> ```
>
> ```python
> from datetime import datetime, timedelta
>
> def find_earliest_time_slot(a_availability, b_availability,
> ↪  meeting_duration):
>     a_availability = [(datetime.strptime(start, '%H:%M'),
>     ↪  datetime.strptime(end, '%H:%M')) for start, end in
>     ↪  a_availability]
>     b_availability = [(datetime.strptime(start, '%H:%M'),
>     ↪  datetime.strptime(end, '%H:%M')) for start, end in
>     ↪  b_availability]
>
>     for a_start, a_end in a_availability:
>         for b_start, b_end in b_availability:
>             latest_start = max(a_start, b_start)
>             earliest_end = min(a_end, b_end)
>
>             if earliest_end - latest_start >=
>             ↪  timedelta(minutes=meeting_duration):
>                 return latest_start.strftime('%H:%M'),
>                 ↪  (latest_start +
>                 ↪  timedelta(minutes=meeting_duration)).strftime('%H:%M')
>
>     return None
> ```
>
> ```
> Use cases:
> Question: A and B want to schedule a 1-hour meeting together.
> ↪  A's availability: 12:00 - 12:30, 13:00 - 13:30, 14:30 -
> ↪  15:30, 17:30 - 18:00. B's availability: 09:00 - 11:00,
> ↪  12:00 - 12:30, 13:00 - 13:30, 15:30 - 16:30, 17:30 -
> ↪  18:00. What time slot works best? (if multiple, choose
> ↪  the earliest one)
> Answer: No time slot works.
> ```
>
> ```
> Solution:
> ```
> ```python
> a_availability = [('12:00', '12:30'), ('13:00', '13:30'),
> ↪  ('14:30', '15:30'), ('17:30', '18:00')]
> b_availability = [('09:00', '11:00'), ('12:00', '12:30'),
> ↪  ('13:00', '13:30'), ('15:30', '16:30'), ('17:30',
> ↪  '18:00')]
> meeting_duration = 60
>
> ret = find_earliest_time_slot(a_availability, b_availability,
> ↪  meeting_duration)
> ans = ret if ret else "No time slot works."
> ```
> ```
> Skip two more questions...
> ```

# E  DATASET CONSTRUCTION

For the "schedule meeting" task, we use the following template to generate the dataset:

```
question_format = """A and B want to schedule a {interval}-hour
↪   meeting together.
A's availability: {A_availability}
B's availability: {B_availability}
What time slot works best? (if multiple, choose the earliest
↪   one)"""
```

where the `interval` is randomly sampled from $\{0.5, 1, 1.5\}$, and the availability of A and B are randomly sampled from 8:00-18:00 with 30 minutes as the granularity. The answer is computed by computing the intersection of the two availability sets and then find the earliest time slot that is at least as long as the meeting duration. If there is no such time slot, we return "No time slot works.".

