# OpenReview forum: "Large Language Models as Tool Makers"
_ICLR.cc/2024/Conference — ICLR 2024 poster_

### Official Review · Reviewer_uR2C · 2023-10-30

**Soundness:** 3 good
**Presentation:** 3 good
**Contribution:** 1 poor
**Rating:** 5
**Confidence:** 4

**Summary:**

The authors present a Tool-Making and Tool-Using framework with two separate LLMs and a functional caching mechanism to re-use tools for functionally analogous requests. With the dichotomy of making and using of tools, the former is assigned to heavier models (e.g. GPT-4) to implement python utility functions while the latter can be conducted with cheaper models (e.g. GPT 3.5). This strategy saves costs and, at the same time, doesn't impact performance as much.

**Strengths:**

The paper was well-written and easy to read. The description of the problem and the proposed solution were clearly explained.

The idea of "dynamically generating tools from a stream of tasks" is novel (to the extent of the reviewer's knowledge) and if explored more concretely and evaluated end-to-end can be a considerable contribution to the community with many applied use cases in the industry.

**Weaknesses:**

The paper seems to be lacking originality and doesn't propose a significant contribution to the LLM domain. Here are the reasons for this conclusion:

Generating Python programs (utility functions), or Tool-making,  has been suggested before in similar works such as that of Chameleon (Lu et al.) referred to there as the "Program Generator" module or the Voyager (Wang et al) or the Creator (Qian et al). Tool-using, as well, is a widely used approach within the realm of LLMs to the extent that OpenAI has released models with function-calling abilities. With these points in mind, the novelty of the proposed work gets limited to using lighter models (cost effectiveness) for tool-using to reduce costs and introducing the function cache (reusibility). We appreciate that the authors have clearly credited the mentioned works with their tool-making ideas, particularly the Chameleon with its Python tools where the "Program Verifier" is the couterpart to the proposed "Tool Verification" step in the current paper.

The paper initially gives the impression that the functional cache and the dispatcher to be playing a central role in the proposed LATM. However, the dispatcher was only adopted in the stream of experiments and, even then, in a set of isolated tests. Accordingly it is not clear if the end-to-end performance of the LATM method across heterogenous tasks was evaluated trusting the reliability of the dispatcher and the functional cache to make the right calls.

**Questions:**

N/A

---

> ### Author Response · Authors · 2023-11-21
> **Acknowledging Reviewer Feedback and Clarifying Key Points**
>
> We appreciate Reviewer uR2C's thoughtful feedback on our work. However, we respectfully disagree with the assessment that our contribution is not novel or significant enough for publication.
>
> The reviewer mentions prior work on tool making, such as Voyager and Creator. As we noted in our related work section, these submissions were developed concurrently and independently of our work. The key novelty of our approach is the integration of tool making and tool using in a unified framework, enabling the strengths of different scale LLMs to be combined in a complementary way. To our knowledge, this specific approach is novel.
>
> We agree that the effectiveness of the dispatcher and functional cache could be evaluated more thoroughly in an end-to-end manner across diverse tasks. Doing this properly would require developing a comprehensive benchmark and methodology for evaluation, which is a challenging and interesting research problem in itself that we plan to investigate in follow-on work. However, given that the core ideas are novel, we feel that our initial feasibility study makes a sufficiently meaningful contribution, especially since it demonstrated the potential benefits of our approach.

---

> > ### Comment · Reviewer_uR2C · 2023-11-22
> > **LLM Tool Creating Literature**
> >
> > Thank you for highlighting the concurrent and independent development of the mentioned similar ideas. The LLM tool creation/utilization is quite new, however, both tool creation nor utliziation have been introduced and explored in similar works (CREATOR by Qian et al, 2023, and more). I re-evaluate the manuscript in light of your response, however, in the context of the available literature and the scope of experiments conducted in the manuscript, the current verdict is valid.

---

### Official Review · Reviewer_hBc8 · 2023-10-31

**Soundness:** 3 good
**Presentation:** 2 fair
**Contribution:** 2 fair
**Rating:** 5
**Confidence:** 5

**Summary:**

Recently, LLMs have demonstrated impressive problem-solving ability to address complex tasks. In this paper, authors introduce a *closed-loop* framework, called LLMs As Tool Makers (LATM). More specifically, this paper introduces two phases: tool making and tool using. Tool making uses LLMs to make new tools via a form of Python utility function and tool using means using tools for problem-solving. Experimental results demonstrate that LATM can achieve performance that matches GPT-4.

**Strengths:**

1. This paper introduces how to use LLMs to make a tool, which sounds interesting. To fulfill this point, this paper proposes two stages: Tool Making and Tool Using. For Tool Making, the authors adopt three sub-stages: Tool Proposing, Tool Verification, and Tool Wrapping to achieve the target.
2. Experimental results demonstrate the proposed method can improve the performance of LLMs in utilizing tools.

**Weaknesses:**

1. Tool making is an interesting idea. However, one weakness in this direction is that we could have many external tools in real-world scenarios and the capability of tool-making should be able to explore new tools that do not have before. Therefore, to further improve the feasibility of the proposed method, it will be better to explore the capability of tool-making to make new tools when also considering the existence of external tools.
2. Compared with Tool making, the design of Tool use is relatively trivial. Many works (e.g., AutoGPT, HuggingGPT) have explored the feasibility of tool use by using LLMs.

**Questions:**

1. This paper mainly utilized the GPT-based models for evaluation, do you try any other open-source LLMs?
2. In Tool verification, the authors utilize 3 validation samples. So how to obtain these validation samples? Are these samples manually created and Are these validation samples fixed or different when meeting different user instructions? What are the generalizations of these examples in different scenarios?
3. Based on my observations, the current LATM can only make tools in language form and cannot support multimodal ability, right?

---

> ### Author Response · Authors · 2023-11-21
> **Acknowledging Reviewer Feedback and Clarifying Key Points**
>
> We appreciate Reviewer hBc8's feedback on our work. We are pleased they found the idea of using LLMs for tool making interesting and novel.
>
> Regarding weakness 1, we agree exploring the tool maker's ability to create entirely new tools beyond existing APIs would be an exciting direction for future work. Our initial study focused on a simpler closed-loop setting, but we plan to investigate more open-ended tool making scenarios as well.
>
> For weakness 2, we agree that tool use with LLMs has been explored in prior work, as cited in our paper. Our contribution is showing tool making can further enhance the performance and efficiency of tool use.
>
> Responding to the reviewer's questions:
>
> Q1) We have not yet evaluated our approach on other open-source LLMs. Testing on other open-source LLMs could illuminate generalizability. However, as our results show, even GPT-4 sometimes fails, so advanced LLMs may be needed for robust tool making. But expanding to other model families is a helpful suggestion for future work.
>
> Q2) In our evaluation, we use the examples from a dev set as validation examples. However, in a streaming setting, validation examples can be collected by using the first few results generated by powerful LLM such as GPT-4 or human-authored.
>
> Q3) Our current implementation focuses on textual tools, yet the method is general. Extending to multimodal abilities is an interesting direction we hope to explore.

---

### Official Review · Reviewer_6T59 · 2023-10-31

**Soundness:** 3 good
**Presentation:** 3 good
**Contribution:** 2 fair
**Rating:** 6
**Confidence:** 3

**Summary:**

The paper introduces the "Large Language Models as Tool Makers (LATM)" framework, which explores the idea of language models creating their own reusable tools for problem-solving. LATM has the ability to craft tools to address challenges. It comprises two phases: tool making and tool using, with a division of labor between powerful and cost-effective models. Additionally, LATM introduces a functional cache mechanism to reduce serving costs. Experiments on a range of complex reasoning tasks show that LATM achieves high performance while being cost-effective.

**Strengths:**

1. The high-level idea of this paper is intuitive and makes sense. It introduces an innovative approach for utilizing large language models, allowing them to create reusable tools, resembling human tool-making abilities.
2. This paper focuses on the practical setting, by strategically allocating the labor between powerful and lightweight models optimize performance while reducing computational costs.
3. The experiments show that the proposed method achieves better performance.

**Weaknesses:**

1. The paper states that they use a self-verification process with a few labeled instances. However, it is not clear whether the performance is stable with a different selection of few-shot validation examples.
2. It is not clear how the tool maker can rectify the incorrect tools made in previous rounds (mentioned in 3.1). More case studies are recommended.
3. Is it possible to evaluate other tool-making benchmarks such as ToolBench[1], and APIBank[2]? These datasets will be useful in faithfully evaluating the tool-making ability of LLMs.

[1] Qin, Yujia, et al. "Tool learning with foundation models." arXiv preprint arXiv:2304.08354 (2023).

[2] Li, Minghao, et al. "Api-bank: A benchmark for tool-augmented llms." arXiv preprint arXiv:2304.08244 (2023).

**Questions:**

1. What is the price for the tool maker module?
2. Can you provide more case studies on how the tool makers conduct the self-verification process?

---

> ### Author Response · Authors · 2023-11-21
> **Acknowledging Reviewer Feedback and Clarifying Key Points**
>
> We thank Reviewer 6T59 for their constructive feedback on our paper. We are pleased the reviewer found the high-level idea intuitive and sees the potential for utilizing large language models as tool makers.
>
> Regarding the stability with different validation examples (W1), we clarify that the self-verification serves only as a simple sanity check. We tried three different sets of validation examples, they can all filter out apparent mistakes and produce correct tools. As the reviewer noted, studying the self-verification process itself is not the focus of this work, and we kindly refer the reviewer to the cited papers in our paper for further discussion.
>
> The suggested ToolBench and APIBank benchmarks (W3) do seem promising for evaluating tool use. However, as we noted, benchmarks focused on workflow customization may be better suited for assessing tool-making abilities. We appreciate these references and will consider evaluating on these datasets in follow-on work once tool-making benchmarks are available.
>
> For Q1, we use GPT-3.5 Turbo as the tool maker module for the main experiments. Its price is 1$/1M tokens for the input and 2$/1M tokens for the output, 30 times cheaper than GPT-4.

---

### Official Review · Reviewer_CctH · 2023-11-01

**Soundness:** 3 good
**Presentation:** 4 excellent
**Contribution:** 1 poor
**Rating:** 3
**Confidence:** 4

**Summary:**

The authors propose a novel method for prompting large language models that instructs them to create their own tools implemented as Python functions which help them solve various tasks that might otherwise be challenging to solve without any tool use. The idea is to prompt a larger LLM (e.g. GPT-4) to write a Python function for solving some given task (along with simple unit tests and documentation with example usage) and then passing this as context to a smaller LLM (e.g. GPT-3.5) and asking it to solve a problem by using a given tool. The key assumption is that larger LLMs are more capable of writing good tool functions compared to smaller LLMs. However, since larger LLMs are more expensive to run, it is beneficial to have them produce a tool that can then be successfully reused by smaller LLMs which are cheaper to run. Additionally, the authors propose a scheme for using another LLM that can decide, given a task, whether it needs to ask the larger LLM to produce a tool, or if it can reuse some of the already produced tools and pass them to the smaller LLM. The authors evaluate their method against reasoning tasks and show the feasibility of their approach.

**Strengths:**

**S1**: The paper presentation is relatively clear and easy to follow. Prior work is adequately referenced.

**S2**: The method is relatively simple which makes it easily applicable. On top of that, the overall approach and distribution of roles between the tool-maker, tool-user, and dispatcher are sensible in this context.

**S3**: The proposed method seems to perform favorably compared to the chain-of-thought baseline across 6 reasoning tasks.

**Weaknesses:**

**W1**: My key issue with the paper is that I'm not sure how substantial the contribution is. I understand the recent enthusiasm surrounding large language models. Furthermore, I do think there could be some scientific merit to studying questions like "What can LLMs do?", "Where do they fail?", and "If we do X, can we make LLMs do Y?" as these questions could deepen our understanding of LLMs, their capabilities, and potential for making them even more powerful. However, prompting LLMs is relatively easy. Hence, to publish a paper about prompting LLMs I would expect either some extremely non-trivial and surprising method or a very wide analysis across a large number of tasks that could further our understanding of the LLMs' capabilities and limitations. On top of that, what I would really love is to see a paper that is able to provide a compelling explanation of why LLMs are able or not able to do something (e.g. solve the Chinese remainder theorem task). On the other hand, a method that essentially says "Hey LLM here is a task, can you write a tool that solves it, etc..." is hardly something I would find surprising, especially given that the authors compare it against only a single baseline (which is equally simple in my opinion) and across only a small handful of relatively simple tasks. PS: I am OK to be challenged on this view (either by the authors and/or by other reviewers and PC members) but this is how I see things. ***TLDR***: In my opinion, the novelty bar for papers that propose methods to prompt LLMs should be much higher and should require much more effort and creativity than is presented in this work.

**Questions:**

**Q1**: Why does CoT achieve a score of 0.0 on the "Chinese Remainder Theorem" problem and LATM achieves 100.0? Also, how does vanilla GPT fare on that task (and why does it fail if it fails)?

---

> ### Author Response · Authors · 2023-11-21
> **Respectfully Disagreeing with Reviewer Assessment of Novelty and Contribution**
>
> We appreciate the reviewer's thoughtful comments on our work. However, we respectfully disagree with the assessment that our contribution is not substantial enough for publication.
>
> The reviewer argues that novelty should require "extremely non-trivial and surprising" methods. While we agree such approaches are valuable, simple but effective techniques also deserve recognition, especially if they *enable new capabilities*. Our approach of using larger LLMs to create tools for smaller LLMs provides a novel and pragmatic way to improve efficiency and task performance. Though conceptually straightforward, this idea was not obvious prior to our work.
>
> We would also like to clarify that our main contribution is not just “prompting LLMs”, instead, we proposed a novel framework to team up multiple language models, and enabled task-solving ability close to the stronger model’s performance while at the weaker models’ serving cost. To the best of our knowledge, such a framework did not exist before and our proposed LATM is a novel way to significantly improve LLMs’ serving efficiency.
>
> We acknowledge the reviewer's point that more extensive evaluation across a diverse range of tasks would further demonstrate the capabilities and limitations of our approach. However, as an initial proof of concept study, our focus was to validate the potential of this novel technique on a small but representative set of reasoning tasks. Note all the tasks we studied in this paper are not “simple tasks”, they are from the Big-Bench Hard subset (i.e., tasks for which prior language model evaluations did not outperform the average human-rater, see https://github.com/suzgunmirac/BIG-Bench-Hard). As we show in the paper, a good model like GPT-3.5 only achieves performance in the range of 20% to 60% on these hard tasks, while our proposed LATM can improve the performance significantly (80% to 100%) while maintaining a similar inference cost. We hope our work stimulates the creation of more comprehensive natural language datasets for human-computer interaction, enabling more thorough evaluation in follow-up studies. For now, our limited but promising results successfully demonstrate the approach's feasibility and set the stage for broader adoption and assessment by the research community.
>
> For Q1, as noted in the paper, we apply direct few-shot prompting without CoT on the Chinese Remainder Theorem task, since readily available demonstrations were unavailable. The failure of GPT-4 is that this task requires many steps of intermediate calculation, which GPT-4 cannot effectively perform internally, even with prompting. Even if we manually add demonstrations with CoT, they would be extremely long, exacerbating context length limitations.
>
> In summary, we believe our approach meaningfully advances the capabilities of LLMs on complex reasoning tasks and also provides novel insights on improving the efficiency of the LLM system.

---

### Meta-Review · Area_Chair_8rRv · 2023-12-14

**Metareview:**

Making a decision on this paper actualizes the complicated question on how to judge pure LLM papers. You might argue that LLM research - basically, studying what LLMs do and inventing new ways of using them - is a new, emerging field. As so often when a new field emerges, it's quite easy to do novel things. There's just a lot of unexplored questions. When reviewing a paper about such an "easy win" for a venue in an established field, it is easy to reject it because research should be hard. Which is... wrong. The funny thing is that ICLR came about partly because papers in the new field of deep learning were seen as not really serious and somehow "too easy" for real ML people.

It seems to me that this is a novel paper with well-executed empirical work. I think it should be accepted.

**Justification For Why Not Higher Score:**

It's not _that_ novel after all.

**Justification For Why Not Lower Score:**

It's good work.

---

### Decision · Program_Chairs · 2024-01-16

Accept (poster)